# Fast Approximation of the Sliced-Wasserstein Distance Using Concentration of Random Projections

**Kimia Nadjahi**[1*],    **Alain Durmus**[2],    **Pierre E. Jacob**[3],
**Roland Badeau**[1],    **Umut Şimşekli**[4]

1: LTCI, Télécom Paris, Institut Polytechnique de Paris, France

2: Université Paris-Saclay, ENS Paris-Saclay, CNRS,
Centre Borelli, F-91190 Gif-sur-Yvette, France

3: Department of Information Systems, Decision Sciences and Statistics,
ESSEC Business School, Cergy, France

4: INRIA - Département d'Informatique de l'École Normale Supérieure,
PSL Research University, Paris, France

## Abstract

The Sliced-Wasserstein distance (SW) is being increasingly used in machine learning applications as an alternative to the Wasserstein distance and offers significant computational and statistical benefits. Since it is defined as an expectation over random projections, SW is commonly approximated by Monte Carlo. We adopt a new perspective to approximate SW by making use of the concentration of measure phenomenon: under mild assumptions, one-dimensional projections of a high-dimensional random vector are approximately Gaussian. Based on this observation, we develop a simple deterministic approximation for SW. Our method does not require sampling a number of random projections, and is therefore both accurate and easy to use compared to the usual Monte Carlo approximation. We derive nonasymptotical guarantees for our approach, and show that the approximation error goes to zero as the dimension increases, under a weak dependence condition on the data distribution. We validate our theoretical findings on synthetic datasets, and illustrate the proposed approximation on a generative modeling problem.

## 1 Introduction

Recent years have witnessed the emergence of numerical methods inspired by optimal transport (OT) to solve machine learning problems. In particular, Wasserstein distances are a core ingredient of OT and define metrics between probability measures. Despite their nice theoretical properties, they are in general computationally expensive in large-scale settings. Several workarounds that scale better to large problems have been developed, and include the Sliced-Wasserstein distance (SW, [1, 2]).

The SW metric is a computationally cheaper alternative to Wasserstein as it exploits the analytical form of the Wasserstein distance between univariate distributions. More precisely, consider two random variables $X$ and $Y$ in $\mathbb{R}^d$ with respective distributions $\mu$ and $\nu$, and denote by $\theta_\sharp^\star \mu, \theta_\sharp^\star \nu$ the univariate distributions of the projections of $X, Y$ along $\theta \in \mathbb{R}^d$. SW then compares $\mu$ and $\nu$ by computing $\mathbb{E}[\mathbf{W}(\theta_\sharp^\star \mu, \theta_\sharp^\star \nu)]$, where the expectation $\mathbb{E}$ is taken with respect to $\theta$ uniformly distributed on the unit sphere, and $\mathbf{W}$ is the Wasserstein distance.

In practice, the expectation is typically estimated by Monte Carlo: one uniformly draws $L$ projection directions $\{\theta_l\}_{l=1}^L$, and approximates SW with $L^{-1} \sum_{i=1}^L \mathbf{W}\big((\theta_l)_\sharp^\star \mu, (\theta_l)_\sharp^\star \nu\big)$. Since the Wasserstein

---

[*]Corresponding author: `kimia.nadjahi@telecom-paris.fr`

distance between univariate distributions can easily be computed in closed-form, this scheme leads to significant computational benefits as compared to the Wasserstein distance, provided that $L$ is not chosen too large. SW has then been successfully applied in several practical tasks, such as classification [3, 4], Bayesian inference [5], the computation of barycenters of measures [1, 6], and implicit generative modeling [7–12]. Besides, SW has been shown to offer nice theoretical properties as well. Indeed, it satisfies the metric axioms [13], the estimators obtained by minimizing SW are asymptotically consistent [14], the convergence in SW is equivalent to the convergence in Wasserstein [14, 15], and even though the sample complexity of Wasserstein grows exponentially with the data dimension [16–18], the sample complexity of SW does not depend on the dimension [19]. However, the latter study also demonstrated with a theoretical error bound, that the quality of the Monte Carlo estimate of SW depends on the number of projections and the variance of the one-dimensional Wasserstein distances [19, Theorem 6]. In other words, to ensure that the induced approximation error is reasonably small, one might need to choose a large value for $L$, which inevitably increases the computational complexity of SW. Alternative approaches have been proposed to overcome this issue, and mainly consist in picking more "informative" projection directions: e.g., SW based on orthogonal projections [10, 20], maximum SW [21], generalized SW distances [22] and distributional SW distances [23].

In this paper, we adopt a different perspective and leverage concentration results on random projections to approximate SW: previous work showed that, under relatively mild conditions, the typical distribution of low-dimensional projections of high-dimensional random variables is close to some Gaussian law [24, 25]. Recently, this phenomenon has been illustrated with a bound in terms of the Wasserstein distance [26]: let $\{X_i\}_{i=1}^d$ be a sequence of real random variables with distribution $\mu_d$, such that $X_1, \ldots, X_d$ are independent with finite fourth-order moments; then, $\mathbb{E}[\mathbf{W}(\theta_\sharp^\star \mu_d, \mathrm{N}_{\mu_d})^2]$ goes to zero as $d$ increases, where $\mathrm{N}_{\mu_d}$ is a univariate Gaussian distribution whose variance depends on $\mu_d$ and the expectation is taken with respect to a Gaussian variable $\theta$. This result has very recently been used to bound the "maximum-sliced distance" between any probability measure and its Gaussian approximation [27]. In our work, we use it to design a novel technique that estimates SW with a simple *deterministic* formula. As opposed to Monte Carlo, our method does not depend on a finite set of random projections, therefore it eliminates the need of tuning the hyperparameter $L$ and can lead to a significant computational time reduction. Besides, our proposal is quite different from the aforementioned variants of SW which consist in selecting "informative" projection directions: these alternatives are defined as optimization problems whose resolution is challenging (e.g., [23, Section 3.2]) and are then computed by finding an approximate solution. This incurs an additional computational cost and estimation error, while our method directly approximates SW (thus, does not define an alternative distance) via simple deterministic operations, does not rely on any hyperparameters, and comes with theoretical guarantees on its induced error.

The important steps to formulate our approximate SW are summarized as follows. We first define an alternative SW whose projection directions are drawn from the same Gaussian distribution as in [26], instead of uniformly on the unit sphere, and establish its relation with the original SW. By combining this property with [26, Theorem 1], we bound the absolute difference between SW applied to any two probability measures $\mu_d$, $\nu_d$ on $\mathbb{R}^d$, and the Wasserstein distance between the univariate Gaussians $\mathrm{N}_{\mu_d}$, $\mathrm{N}_{\nu_d}$. Then, we explain why the mean parameters of $\mu_d$ and $\nu_d$ should be zero for the approximation error to decrease as $d$ grows. Nevertheless, we show that it is not a limiting factor, by exploiting the following decomposition of SW: SW between $\mu_d$, $\nu_d$ can be equivalently written as the sum of the *difference between their means* and the SW between the *centered versions* of $\mu_d$, $\nu_d$.

Our approach then consists in estimating SW between the centered versions with the Wasserstein term between Gaussian approximations to meet the zero-means condition, and recover SW between the original measures via the aforementioned property. Since the Wasserstein distance between Gaussian distributions admits a closed-form solution, our approximate SW is very easy to compute, and faster than the Monte Carlo estimate obtained with a large number of projections. We derive nonasymptotical guarantees on the error induced by our approach. Specifically, we define a weak dependence condition under which the error is shown to go to zero with increasing $d$. Our theoretical results are then validated with experiments conducted on synthetic data. Finally, we leverage our theoretical insights to design a novel adversarial framework for a typical generative modeling problem in machine learning, and illustrate its advantages in terms of accuracy and computational time, over

generative models based on the Monte Carlo estimate of SW. Our empirical results can be reproduced with our open source code[2].

## 2 Background

We first give some background on optimal transport distances and concentration of measure for random projections. All random variables are defined on a probability space $(\Omega, \mathcal{F}, \mathbb{P})$ with associated expectation operator $\mathbb{E}$. We denote $\mathbb{N}^* = \mathbb{N} \setminus \{0\}$, and for $d \in \mathbb{N}^*$, $\mathcal{P}(\mathbb{R}^d)$ is the set of probability measures on $\mathbb{R}^d$.

### 2.1 Optimal transport distances

Let $p \in [1, +\infty)$ and $\mathcal{P}_p(\mathbb{R}^d) = \left\{ \mu \in \mathcal{P}(\mathbb{R}^d) \ : \ \int_{\mathbb{R}^d} \|x\|^p \, \mathrm{d}\mu(x) < +\infty \right\}$ be the set of probability measures on $\mathbb{R}^d$ with finite moment of order $p$. The Wasserstein distance of order $p$ between any $\mu, \nu \in \mathcal{P}_p(\mathbb{R}^d)$ is defined as

$$\mathbf{W}_p^p(\mu, \nu) = \inf_{\pi \in \Pi(\mu, \nu)} \int_{\mathbb{R}^d \times \mathbb{R}^d} \|x - y\|^p \, \mathrm{d}\pi(x, y) \,, \tag{1}$$

where $\| \cdot \|$ denotes the Euclidean norm, and $\Pi(\mu, \nu)$ the set of probability measures on $\mathbb{R}^d \times \mathbb{R}^d$ whose marginals with respect to the first and second variables are given by $\mu$ and $\nu$ respectively. In some particular settings, $\mathbf{W}_p$ is relatively easy to compute since the optimization problem in (1) admits a closed-form solution: we give two examples that will be useful in the rest of the paper.

**Gaussian distributions.** Denote by $\mathrm{N}(\mathbf{m}, \Sigma)$ the Gaussian distribution on $\mathbb{R}^d$ with mean $\mathbf{m} \in \mathbb{R}^d$ and covariance matrix $\Sigma \in \mathbb{R}^{d \times d}$ symmetric positive-definite. The Wasserstein distance between two Gaussian distributions, also known as the Wasserstein-Bures metric, is given by [28]

$$\mathbf{W}_2^2\{\mathrm{N}(\mathbf{m}_1, \Sigma_1), \mathrm{N}(\mathbf{m}_2, \Sigma_2)\} = \|\mathbf{m}_1 - \mathbf{m}_2\|^2 + \mathrm{Tr}\left[\Sigma_1 + \Sigma_2 - 2\left(\Sigma_1^{1/2} \Sigma_2 \Sigma_1^{1/2}\right)^{1/2}\right] \,, \tag{2}$$

where $\mathrm{Tr}$ is the trace operator.

**Univariate distributions.** Consider $\mu, \nu \in \mathcal{P}_p(\mathbb{R})$, and denote by $F_\mu^{-1}$ and $F_\nu^{-1}$ the quantile functions of $\mu$ and $\nu$ respectively. By [29, Theorem 3.1.2.(a)],

$$\mathbf{W}_p^p(\mu, \nu) = \int_0^1 \left| F_\mu^{-1}(t) - F_\nu^{-1}(t) \right|^p \, \mathrm{d}t \,. \tag{3}$$

If $\mu = n^{-1} \sum_{i=1}^n \delta_{x_i}$ and $\nu = n^{-1} \sum_{i=1}^n \delta_{y_i}$, with $\{x_i\}_{i=1}^n, \{y_i\}_{i=1}^n \subset \mathbb{R}^n$ and $\delta_z$ the Dirac distribution with mass on $z$, (3) can simply be calculated by sorting $\{x_i\}_{i=1}^n$ and $\{y_i\}_{i=1}^n$ as $x_{(1)} \leq \ldots \leq x_{(n)}$ and $y_{(1)} \leq \ldots \leq y_{(n)}$. Indeed, in this case, $\mathbf{W}_p^p(\mu, \nu) = n^{-1} \sum_{i=1}^n |x_{(i)} - y_{(i)}|^p$. However, when the empirical distributions are multivariate, $\mathbf{W}_p(\mu, \nu)$ is not analytically available in general, so its computation is expensive: the standard methods used to solve the linear program in (1) have a worst-case computational complexity in $\mathcal{O}(n^3 \log n)$, and tend to have a super-cubic cost in practice [30, Chapter 3].

The Sliced-Wasserstein distance [1, 2] defines a practical alternative metric by leveraging the computational efficiency of $\mathbf{W}_p$ for univariate distributions. Let $\mathbb{S}^{d-1}$ be the $d$-dimensional unit sphere and $\boldsymbol{\sigma}$ the uniform distribution on $\mathbb{S}^{d-1}$. For $\theta \in \mathbb{S}^{d-1}$, $\theta^\star : \mathbb{R}^d \to \mathbb{R}$ denotes the linear form $x \mapsto \langle \theta, x \rangle$ with $\langle \cdot, \cdot \rangle$ the Euclidean inner-product. Then, SW of order $p \in [1, \infty)$ between $\mu, \nu \in \mathcal{P}_p(\mathbb{R}^d)$ is

$$\mathbf{SW}_p^p(\mu, \nu) = \int_{\mathbb{S}^{d-1}} \mathbf{W}_p^p(\theta_\sharp^\star \mu, \theta_\sharp^\star \nu) \mathrm{d}\boldsymbol{\sigma}(\theta) \,, \tag{4}$$

where for any measurable function $f : \mathbb{R}^d \to \mathbb{R}$ and $\xi \in \mathcal{P}(\mathbb{R}^d)$, $f_\sharp \xi$ is the push-forward measure of $\xi$ by $f$: for any Borel set $\mathsf{A}$ in $\mathbb{R}$, $f_\sharp \zeta(\mathsf{A}) = \zeta(f^{-1}(\mathsf{A}))$, with $f^{-1}(\mathsf{A}) = \{x \in \mathbb{R}^d \ : \ f(x) \in \mathsf{A}\}$.

Since $\theta_\sharp^\star \mu, \theta_\sharp^\star \nu$ are univariate distributions, the Wasserstein distances in (4) are conveniently computed using (3). Besides, in practical applications, the expected value in (4) is typically approximated with a standard Monte Carlo method:

---

[2]See `https://github.com/kimiandj/fast_sw`

$$\mathbf{SW}_{p,L}^p(\mu,\nu) = (1/L) \sum_{l=1}^{L} \mathbf{W}_p^p\big((\theta_l)_\sharp^\star\mu, (\theta_l)_\sharp^\star\nu\big), \text{ with } \{\theta_l\}_{l=1}^L \text{ i.i.d. from } \boldsymbol{\sigma} . \tag{5}$$

Computing $\mathbf{SW}_{p,L}$ between two empirical distributions then amounts to projecting sets of $n$ observations in $\mathbb{R}^d$ along $L$ directions, and sorting the projected data. The resulting computational complexity is $\mathcal{O}(Ldn + Ln\log n)$, which is more efficient than $\mathbf{W}_p$ in general. This complexity means that the Monte Carlo estimate is more expensive when $d$, $n$ and $L$ increase, and it is often unclear how $L$ should be chosen in order to control the approximation error; see [19, Theorem 6].

## 2.2 Central limit theorems for random projections

There is a rich literature on the typical behavior of one-dimensional random projections of high-dimensional vectors. To be more specific, let $(\theta_i)_{i\in\mathbb{N}^*}$ be i.i.d. standard one-dimensional Gaussian random variables and $(X_i)_{i\in\mathbb{N}^*}$ be a sequence of one-dimensional random variables. Denote for any $d \in \mathbb{N}^*$, $\theta_{1:d} = \{\theta_i\}_{i=1}^d$ and $X_{1:d} = \{X_i\}_{i=1}^d$. Several central limits theorems ensure that, under relatively mild conditions, the sequence of distributions of $d^{-1/2}\langle\theta_{1:d}, X_{1:d}\rangle \in \mathbb{R}$ given $\theta_{1:d} \in \mathbb{R}^d$ converges in distribution to a Gaussian random variable in probability. This line of work goes back to [24, 25], whose contributions have then been sharpened and generalized in [31–38]. In particular, a recent study [26] gives a quantitative version of this phenomenon. More precisely, denote for any $d \in \mathbb{N}^*$ by $\mu_d^X$, the distribution of $X_{1:d}$ (i.e., the joint distribution of $X_1, X_2, \ldots, X_d$) and $\boldsymbol{\gamma}_d$ the zero-mean Gaussian distribution with covariance matrix $(1/d)\mathbf{I}_d$. Assume that for any $d \in \mathbb{N}^*$, $\mu_d^X \in \mathcal{P}_2(\mathbb{R}^d)$. Then, [26, Theorem 1] shows that there exists a universal constant $C \geq 0$ such that

$$\int_{\mathbb{R}^d} \mathbf{W}_2^2\big(\theta_\sharp^\star\mu_d^X, \mathrm{N}\big(0, d^{-1}\mathtt{m}_2(\mu_d^X)\big)\big)\,\mathrm{d}\boldsymbol{\gamma}_d(\theta) \leq C\Xi_d(\mu_d^X) , \text{ with} \tag{6}$$

$$\Xi_d(\mu_d^X) = d^{-1}\{\alpha(\mu_d^X) + \big(\mathtt{m}_2(\mu_d^X)\beta_1(\mu_d^X)\big)^{1/2} + \mathtt{m}_2(\mu_d^X)^{1/5}\beta_2(\mu_d^X)^{4/5}\} , \tag{7}$$

$$\mathtt{m}_2(\mu_d^X) = \mathbb{E}\left[\|X_{1:d}\|^2\right] , \quad \alpha(\mu_d^X) = \mathbb{E}\left[\left|\|X_{1:d}\|^2 - \mathtt{m}_2(\mu_d^X)\right|\right] , \quad \beta_q(\mu_d^X) = \mathbb{E}^{\frac{1}{q}}\big[|\langle X_{1:d}, X'_{1:d}\rangle|^q\big] , \tag{8}$$

where $q \in \{1, 2\}$ and $(X'_i)_{i\in\mathbb{N}^*}$ is an independent copy of $(X_i)_{i\in\mathbb{N}^*}$. A formal statement of this result is also given for completeness in the supplement.

# 3 Approximate Sliced-Wasserstein distance based on concentration of random projections

We develop a novel method to approximate the Sliced-Wasserstein distance of order 2, by extending the bound in (6) and deriving novel properties for SW. We then derive nonasymptotical guarantees of the corresponding approximation error, which ensure that our estimate is accurate for high-dimensional data under a weak dependence condition.

## 3.1 Sliced-Wasserstein distance with Gaussian projections

First, to enable the use of (6) for the analysis of SW, we introduce a variant of $\mathbf{SW}_p$ (4) whose projections are drawn from the Gaussian distribution considered in (6), instead of uniformly on the sphere. The Sliced-Wasserstein distance of order $p \in [1, +\infty)$ based on Gaussian projections is defined for any $\mu, \nu \in \mathcal{P}_p(\mathbb{R}^d)$ as

$$\widetilde{\mathbf{SW}}_p^p(\mu,\nu) = \int_{\mathbb{R}^d} \mathbf{W}_p^p(\theta_\sharp^\star\mu, \theta_\sharp^\star\nu)\mathrm{d}\boldsymbol{\gamma}_d(\theta) . \tag{9}$$

In the next proposition, we establish a simple mathematical relation between traditional SW and the newly introduced one: we prove that $\widetilde{\mathbf{SW}}_p$ is equal to $\mathbf{SW}_p$ up to a proportionality constant that only depends on the data dimension $d$ and the order $p$.

**Proposition 1.** *Let $p \in [1, +\infty)$. Then, $\widetilde{\mathbf{SW}}_p$ (9) is related to $\mathbf{SW}_p$ (4) as follows: for any $\mu, \nu \in \mathcal{P}_p(\mathbb{R}^d)$, $\widetilde{\mathbf{SW}}_p(\mu,\nu) = (2/d)^{1/2}\left\{\Gamma(d/2 + p/2) / \Gamma(d/2)\right\}^{1/p}\mathbf{SW}_p(\mu,\nu)$, where $\Gamma$ is the Gamma function.*

Since (6) only applies to the Wasserstein distance of order 2, we will focus on SW of that same order in the rest of the paper. In this case, SW with Gaussian projections is *equal* to the original SW. Indeed, we can show that the constant $(2/d)^{1/2} \{\Gamma(d/2 + p/2) / \Gamma(d/2)\}^{1/p}$ defined in Proposition 1 is equal to 1 when $p = 2$, by using the property $\Gamma(d/2 + 1) = (d/2)\Gamma(d/2)$.

### 3.2 Approximate Sliced-Wasserstein distance

Our next result is an easy consequence of (6) and Proposition 1, and shows that the absolute difference between $\mathbf{SW}_2(\mu_d, \nu_d)$ and $\mathbf{W}_2\{\mathrm{N}(0, d^{-1}\mathtt{m}_2(\mu_d)), \mathrm{N}(0, d^{-1}\mathtt{m}_2(\nu_d))\}$ for any $\mu_d, \nu_d \in \mathcal{P}_2(\mathbb{R}^d)$, is bounded from above by $\Xi_d(\mu_d) + \Xi_d(\nu_d)$ (7).

**Theorem 1.** *There exists a universal constant $C > 0$ such that for any $\mu_d, \nu_d \in \mathcal{P}_2(\mathbb{R}^d)$,*

$$\left| \mathbf{SW}_2(\mu_d, \nu_d) - \mathbf{W}_2\{\mathrm{N}(0, d^{-1}\mathtt{m}_2(\mu_d)), \mathrm{N}(0, d^{-1}\mathtt{m}_2(\nu_d))\} \right| \leq C\big(\Xi_d(\mu_d) + \Xi_d(\nu_d)\big)^{1/2} , \quad (10)$$

*where, for $\xi_d \in \{\mu_d, \nu_d\}$, $\Xi_d(\xi_d)$ and $\mathtt{m}_2(\xi_d)$ are defined in (7) and (8) respectively.*

Since $\mathbf{W}_2\{\mathrm{N}(0, d^{-1}\mathtt{m}_2(\mu_d)), \mathrm{N}(0, d^{-1}\mathtt{m}_2(\nu_d))\}$ has a closed-form solution by (2), it provides a computationally efficient approximation of $\mathbf{SW}_2(\mu_d, \nu_d)$ whose accuracy is quantified by Theorem 1. Next, we identify settings where this approximation is accurate, by analyzing the error $\Xi_d(\mu_d) + \Xi_d(\nu_d)$.

Our first observation is that $\mu_d$ and $\nu_d$ should have zero means for the error to go to zero as $d \to +\infty$, and we develop a novel approximation of SW that takes into account this constraint. Going back to the definition of $\Xi_d(\mu_d^X)$ in (7), setting $\bar{X}_i = X_i - \mathbb{E}[X_i]$ and $\bar{X}_i' = X_i' - \mathbb{E}[X_i']$, we get

$$\mathtt{m}_2(\mu_d^X) = \mathbb{E}[\|\bar{X}_{1:d}\|^2] + \|\mathbb{E}[X_{1:d}]\|^2 \tag{11}$$

$$\beta_2^2(\mu_d^X) = \mathbb{E}\left[\langle \bar{X}_{1:d}, \bar{X}_{1:d}' \rangle^2\right] + 4\mathbb{E}\left[\langle \mathbb{E}[X_{1:d}], \bar{X}_{1:d} \rangle^2\right] + \|\mathbb{E}[X_{1:d}]\|^4 . \tag{12}$$

By Equations (11) and (12), since in practice the norm of the mean $\mathbb{E}[X_{1:d}]$ is expected to increase linearly with $d^{1/2}$ at least, so are $\mathtt{m}_2(\mu_d^X)$ and $\beta_2(\mu_d^X)$ as functions of $d$. As a consequence, $\Xi_d(\mu_d^X)$ cannot be shown to converge to 0 as $d \to \infty$ in this setting, but only to be bounded. However, if the data are centered, the norm of the mean is zero, thus $\Xi_d(\mu_d^X)$ might be decreasing. Therefore, we derive a convenient formula to compute $\mathbf{SW}_2(\mu_d, \nu_d)$ from $\mathbf{SW}_2(\bar{\mu}_d, \bar{\nu}_d)$ where for any $\xi_d \in \mathcal{P}_2(\mathbb{R}^d)$, $\bar{\xi}_d$ is the centered version of $\xi$, *i.e.* the pushforward measure of $\xi_d$ by $x \mapsto x - \mathtt{m}_{\xi_d}$ with $\mathtt{m}_{\xi_d} = \int_{\mathbb{R}^d} y \, \mathrm{d}\xi_d(y)$. This result is the last ingredient to formulate our approximation of SW.

**Proposition 2.** *Let $\mu_d, \nu_d \in \mathcal{P}_2(\mathbb{R}^d)$ with respective means $\mathbf{m}_{\mu_d}, \mathbf{m}_{\nu_d}$. Then, the Sliced-Wasserstein distance of order 2 can be decomposed as*

$$\mathbf{SW}_2^2(\mu_d, \nu_d) = \mathbf{SW}_2^2(\bar{\mu}_d, \bar{\nu}_d) + (1/d)\|\mathbf{m}_{\mu_d} - \mathbf{m}_{\nu_d}\|^2 . \tag{13}$$

Based on Proposition 2, instead of estimating $\mathbf{SW}_2(\mu_d, \nu_d)$ with $\mathbf{W}_2\{\mathrm{N}(0, d^{-1}\mathtt{m}_2(\mu_d)), \mathrm{N}(0, d^{-1}\mathtt{m}_2(\nu_d))\}$ directly, we propose approximating $\mathbf{SW}_2(\bar{\mu}_d, \bar{\nu}_d)$ with $\mathbf{W}_2\{\mathrm{N}(0, d^{-1}\mathtt{m}_2(\bar{\mu}_d)), \mathrm{N}(0, d^{-1}\mathtt{m}_2(\bar{\nu}_d))\}$ and then using (13). This strategy yields our final approximation of SW, which is defined for any $\mu_d, \nu_d \in \mathcal{P}_2(\mathbb{R}^d)$ as

$$\widehat{\mathbf{SW}}_2^2(\mu_d, \nu_d) = \mathbf{W}_2^2\{\mathrm{N}(0, d^{-1}\mathtt{m}_2(\bar{\mu}_d)), \mathrm{N}(0, d^{-1}\mathtt{m}_2(\bar{\nu}_d))\} + (1/d)\|\mathbf{m}_{\mu_d} - \mathbf{m}_{\nu_d}\|^2 , \tag{14}$$

where for $\xi_d \in \{\bar{\mu}_d, \bar{\nu}_d\}$, $\mathtt{m}_2(\xi_d)$ is defined in (8). Note that (14) can be simplified since by (2), $\mathbf{W}_2^2\{\mathrm{N}(0, d^{-1}\mathtt{m}_2(\bar{\mu}_d)), \mathrm{N}(0, d^{-1}\mathtt{m}_2(\bar{\nu}_d))\} = d^{-1}(\mathtt{m}_2(\bar{\mu}_d)^{1/2} - \mathtt{m}_2(\bar{\nu}_d)^{1/2})^2$. Besides, if $\mu_d$ and $\nu_d$ are both supported on a finite set of points, $\widehat{\mathbf{SW}}_2(\mu_d, \nu_d)$ has a closed-form expression: given $\xi_d = n^{-1} \sum_{j=1}^{n} \delta_{x^{(j)}} \in \mathcal{P}_2(\mathbb{R}^d)$ with $x^{(j)} \in \mathbb{R}^d$ for $j \in \{1, \ldots, n\}$, we then have $\mathbf{m}_{\xi_d} = n^{-1} \sum_{j=1}^{n} x^{(j)}$, and $\mathtt{m}_2(\xi_d) = n^{-1} \sum_{j=1}^{n} \|x^{(j)}\|^2$. The associated computational complexity is therefore in $\mathcal{O}(dn)$.

Hence, we introduced an alternative technique to estimate SW which does not rely on a finite set of random projections, as opposed to the commonly used Monte Carlo technique (5). Our approach thus eliminates the need for practitioners to tune the number of projections $L$, but also to sort the projected data. As a consequence, it is more efficient to compute $\widehat{\mathbf{SW}}_2(\mu_d, \nu_d)$ than $\mathbf{SW}_{2,L}(\mu_d, \nu_d)$ for large $L$. We illustrate this latter point with empirical results in Section 4.

### 3.3 Error analysis under weak dependence

We have discussed why centering the data is important to ensure that the approximation error goes to zero with increasing $d$. Next, we introduce a weak dependence condition under which the error is guaranteed to decrease as $d$ increases.

We first consider a setting mentioned in [26] where $\mu_d = \mu^{(1)} \otimes \cdots \otimes \mu^{(d)}$ and $\nu_d = \nu^{(1)} \otimes \cdots \otimes \nu^{(d)}$, $\otimes$ denoting the tensor product of measures, and $\mu^{(j)}, \nu^{(j)} \in \mathcal{P}_4(\mathbb{R})$ for $j \in \{1, \ldots, d\}$. We prove in this case that $\mathbf{W}_2\{\mathrm{N}(0, d^{-1}\mathrm{m}_2(\mu_d)), \mathrm{N}(0, d^{-1}\mathrm{m}_2(\nu_d))\}$ converges to $\mathbf{SW}_2(\mu_d, \nu_d)$ at a rate of $d^{-1/8}$. This result is reported in the supplementary document, and can be interpreted as an extension of [26, Corollary 3] for SW.

We emphasize that the assumptions of this first setting severely restrict the scope of application of our approximation method: in several statistical and machine learning tasks, the random variables of interest $\{X_i\}_{i=1}^d$ are not independent from each other (e.g. for image data, each $X_i$ typically represents the value of a pixel at a certain position, thus depends on the neighboring pixels). Therefore, we relax this independence condition by considering a concept of 'weak dependence' inspired by [39] and properly defined in Definition 1.

**Definition 1.** *Let $(X_j)_{j \in \mathbb{N}^*}$ be a stationary sequence of one-dimensional random variables with mean zero, i.e. $X_i$ and $X_j$ have the same distribution for any $i, j \in \mathbb{N}^*$ and $\mathbb{E}[X_1] = 0$. We say that $(X_j)_{j \in \mathbb{N}^*}$ is* fourth-order weakly dependent *if there exist some constant $K \geq 0$ and a nonincreasing sequence of real coefficients $\{\rho(n)\}_{n \in \mathbb{N}}$ such that, for any $i, j \in \mathbb{N}^*$, $i \leq j$,*

$$|\mathrm{Cov}(X_i^2, X_j^2)| \leq K\rho(j-i), \qquad |\mathrm{Cov}(X_i, X_j)| \leq K\rho(j-i). \tag{15}$$

*In addition, the sequence $\{\rho(n)\}_{n \in \mathbb{N}}$ satisfies $\sum_{n=0}^{+\infty} \rho(n) \leq \rho_\infty < +\infty$.*

Intuitively, in practical applications, the weak dependence condition would essentially require the components of the observations not to exhibit strong correlations; yet, they are allowed to depend on each other. Furthermore, since our weak dependence condition is weaker than the one introduced in [39, Theorem 1], it is satisfied by the various examples of models described in [39, Section 5]. We present some of them below, to illustrate Definition 1 more clearly.

1) *Gaussian processes* and *associated processes* [40, Section 3.1], provided that they are stationary.
2) *Bernoulli shifts*: $X_t = H(\varepsilon_t, \ldots, \varepsilon_{t-r})$ for $t \in \mathbb{N}^*$, where $H : \mathbb{R}^{r+1} \to \mathbb{R}$ is a measurable function and $(\varepsilon_i)_{i \in \mathbb{N}^*}$ is a sequence of i.i.d. real random variables. A simple example of such process is given by *moving-average models*.
3) *Autoregressive models*, defined as $X_t = f(X_{t-1}, \ldots, X_{t-r}) + \varepsilon_t$ for $t \in \mathbb{N}^*$, where $(\varepsilon_i)_{i \in \mathbb{N}^*}$ a sequence of i.i.d. real random variables with $\mathbb{E}|\varepsilon_1| < \infty$, and $|f(u_1, \ldots, u_r) - f(v_1, \ldots, v_r)| \leq \sum_{i=1}^r a_i |u_i - v_i|$ for some $a_1, \ldots, a_r \geq 0$ such that $\left(\sum_{i=1}^r a_i\right)^{1/r} < 1$.

We then consider a sequence of fourth-order weakly dependent random variables $(X_j)_{j \in \mathbb{N}^*}$, and prove that $\Xi_d(\mu_d^X)$ goes to zero as $d \to \infty$, with a rate of convergence depending on $\{\rho(n)\}_{n \in \mathbb{N}}$. This result is given in the supplementary document, and helps us refine Theorem 1 under this weak dependence condition: the next corollary establishes that the error approaches 0 at a rate of $d^{-1/8}$.

**Corollary 1.** *Let $(X_j)_{j \in \mathbb{N}^*}$ and $(Y_j)_{j \in \mathbb{N}^*}$ be sequences of random variables which are fourth-order weakly dependent. Set for any $d \in \mathbb{N}^*$, $X_{1:d} = \{X_j\}_{j=1}^d$ and $Y_{1:d} = \{Y_j\}_{j=1}^d$, and denote by $\mu_d$, $\nu_d$ the distributions of $X_{1:d}$, $Y_{1:d}$ respectively. Then, there exists a universal constant $C > 0$ such that $\left|\mathbf{SW}_2(\mu_d, \nu_d) - \mathbf{W}_2(\mathrm{N}(0, d^{-1}\mathrm{m}_2(\mu_d)), \mathrm{N}(0, d^{-1}\mathrm{m}_2(\nu_d)))\right| \leq Cd^{-1/8}$.*

Hence, by replacing the independence condition of the first setting with weak dependence, we broaden the scope of application whilst guaranteeing that the approximation error goes to zero as $d$ increases. We finally note that in these two settings, the data are required to have zero mean, which is automatically verified with our approximation method since we estimate SW between the centered distributions (see eq. (14)).

## 4 Experiments

**Synthetic experiments.** The goal of these experiments is to illustrate our theoretical results derived in Section 3. In each setting, we generate two sets of $d$-dimensional samples, denoted by $\{x^{(j)}\}_{j=1}^n$

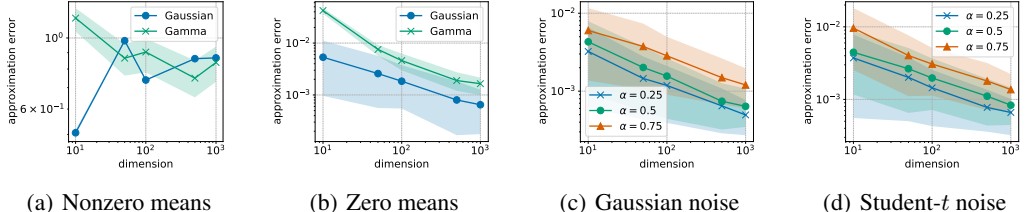

| (a) Nonzero means | (b) Zero means | (c) Gaussian noise | (d) Student-$t$ noise |

Figure 1: Analysis of the approximation according to the dimension: in Figures 1(a) and 1(b), data have independent components; in Figures 1(c) and 1(d), they are stationary AR(1) processes. Errors are averaged over 100 runs and reported on log-log scale with their 10th-90th percentiles.

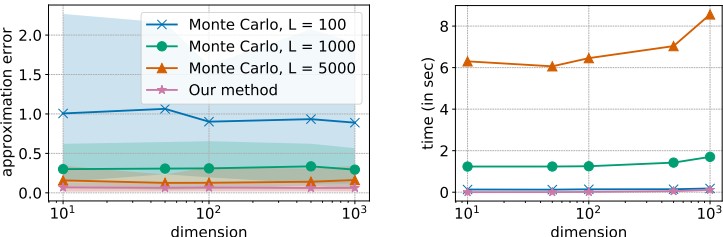

Figure 2: Comparison of different methods to approximate SW, according to their accuracy (left) and computation time (right). The datasets contain $n$ samples of dimension $d$ independently drawn from Gamma distributions, with $d \in [10^1, 10^3]$ and $n = 10^4$. Results are averaged over 100 runs.

and $\{y^{(j)}\}_{j=1}^n$ with $n = 10^4$ and $x^{(j)}, y^{(j)} \in \mathbb{R}^d$ for $j \in \{1, \ldots, n\}$. We then approximate SW between their empirical distributions in $\mathcal{P}_2(\mathbb{R}^d)$, given by $\mu_d = n^{-1} \sum_{j=1}^n \delta_{x^{(j)}}$ and $\nu_d = n^{-1} \sum_{j=1}^n \delta_{y^{(j)}}$.

First, we analyze the consequences of centering data. Here, $\{x^{(j)}\}_{j=1}^n, \{y^{(j)}\}_{j=1}^n$ are $n$ independent samples from Gaussian or Gamma distributions: see the supplementary document for more details. We compute $|\mathbf{W}_2\{\mathrm{N}(0, d^{-1}\mathtt{m}_2(\mu_d)), \mathrm{N}(0, d^{-1}\mathtt{m}_2(\nu_d))\} - \mathbf{SW}_2(\mu_d, \nu_d)|$ on the one hand, and $|\mathbf{W}_2\{\mathrm{N}(0, d^{-1}\mathtt{m}_2(\bar{\mu}_d)), \mathrm{N}(0, d^{-1}\mathtt{m}_2(\bar{\nu}_d))\} - \mathbf{SW}_2(\bar{\mu}_d, \bar{\nu}_d)|$ on the other hand. In the Gaussian case, the exact value of $\mathbf{SW}_2$ is known (we report it in the supplementary document), while for the Gamma distributions, it is approximated with Monte Carlo based on $2 \times 10^4$ random projections. Figures 1(a) and 1(b) show that the error goes to zero as $d$ increases if the data are centered. This confirms our analysis provided in Section 3.2 about the influence of the mean, and in Section 3.3 on sequences of independent random variables.

Next, we consider autoregressive processes of order one (AR(1)). An AR(1) process is defined as $X_1 = \varepsilon_1$ and, for $t \in \mathbb{N}^*$, $X_t = \alpha X_{t-1} + \varepsilon_t$, where $\alpha \in [0, 1]$ and $(\varepsilon_i)_{i \in \mathbb{N}^*}$ is an i.i.d. sequence of real random variables with $\mathbb{E}[\varepsilon_1] = 0$ and finite second-order moment. If $\alpha < 1$, the process has a stationary distribution and $(X_j)_{j \in \mathbb{N}^*}$ satisfies the weak dependence condition in its stationary regime [41]. In practice, we generate a sample by using this recursion formula for $10^4 + d$ steps, and keeping the last $d$ samples. The discarded samples correspond to a "burn-in" phase which helps reaching the stationary solution of the process. We generate $\{x^{(j)}\}_{j=1}^n$ and $\{y^{(j)}\}_{j=1}^n$ using the same distribution for the noise (either a Gaussian or Student's $t$-distribution, as described in the supplementary document). This means that both datasets come from the same distribution, thus the exact value of $\mathbf{SW}_2$ is zero. We plot on Figures 1(c) and 1(d) the approximation error according to $d \in [10, 10^3]$ for different values of $\alpha$. The error converges to zero with increasing $d$, which is consistent with Corollary 1.

Note that Figure 1 exhibits rate of convergence that are better than the one in $d^{-1/8}$ derived in Section 3.3: in Figure 1(b), the slope is approximately $-0.45$ (Gaussian) and $-0.7$ (Gamma), and in Figures 1(c) and 1(d), it is on average $-0.35$. This suggests that our theoretical bounds might be improved, and we further investigate this aspect for the Gaussian case: we consider the case where $\{x^{(j)}\}_{j=1}^n, \{y^{(j)}\}_{j=1}^n$ are $n$ independent samples from Gaussian distributions with diagonal

| Dataset | Model | FID | $T_{\mathbf{SW}}$ (s/epoch) | | $T_{\mathbf{tot}}$ (s/epoch) | |
| | | | GPU | CPU | GPU | CPU |
|---|---|---|---|---|---|---|
| MNIST | SWG | $22.41 \pm 2.34$ | 1.3 | $1.4 \times 10^2$ | 4.5 | $2.7 \times 10^2$ |
| | Reg-SWG | $15.53 \pm 0.88$ | 1.1 | $1.1 \times 10^2$ | 6.5 | $3.0 \times 10^2$ |
| | Reg-det-SWG | $15.72 \pm 0.57$ | 0.07 | 0.2 | 5.3 | $1.5 \times 10^2$ |
| CelebA | SWG | $31.04 \pm 2.78$ | 10.1 | $2.7 \times 10^3$ | $3.9 \times 10^2$ | $1.6 \times 10^4$ |
| | Reg-SWG | $24.14 \pm 0.48$ | 10.0 | $2.7 \times 10^3$ | $4.4 \times 10^2$ | $2.0 \times 10^4$ |
| | Reg-det-SWG | $23.65 \pm 0.93$ | 1.3 | 2.6 | $4.2 \times 10^2$ | $1.7 \times 10^4$ |

Table 1: Results obtained after training generative models on MNIST and CelebA, averaged over 5 runs. FID are reported with their standard deviation (the lower FID, the better). $T_{\mathrm{SW}}$ denotes the average time per epoch for approximating SW. $T_{\mathrm{tot}}$ is the average running time per epoch.

covariance matrices, and we prove that $\mathbb{E}|\mathbf{W}_2\{\mathrm{N}(0, d^{-1}\mathtt{m}_2(\bar{\mu}_d)), \mathrm{N}(0, d^{-1}\mathtt{m}_2(\bar{\nu}_d))\} - \mathbf{SW}_2(\bar{\mu}_d, \bar{\nu}_d)|$ goes to 0 as $dn \to +\infty$ with a convergence rate in $d^{-1/2}n^{-1/2}$. We provide the complete statement and formal proof in Section S3.1 (Proposition S3). This result is consistent with Figure 1(b), and is a first encouraging step towards the following research direction: we will study if our proofs and the ones in [26] can be refined when assuming additional structure on the distributions (e.g., sub-Gaussian and sub-exponential), in order to identify the settings under which our current bounds are tight or can be improved.

Finally, we compare our approximation scheme against the standard Monte Carlo estimation, in terms of accuracy and computation time. We use the same setting as in Figure 1(b), where the $n$ samples are independently drawn from Gamma distributions. We compute $\widehat{\mathbf{SW}}_2(\mu_d, \nu_d)$ (14) and $\mathbf{SW}_{2,L}(\mu_d, \nu_d)$ (5) with $L \in \{100, 1000, 5000\}$, and we compare each approximation with $\mathbf{SW}_{2,2\times10^4}(\mu_d, \nu_d)$, which we consider as the exact value of SW. Figure 2 reports the approximation error and computation time of each scheme for $d \in [10, 10^4]$, and shows that our method is more accurate and faster than Monte Carlo. In particular, when $d = 10^3$, the average computation time of our technique is 0.02s, while the second best approximation (Monte Carlo with $L = 5000$) takes more than 8s. Besides, we observe that Monte Carlo is very sensitive to the hyperparameters, since it loses accuracy when $L$ decreases and gets slower as $L$ and $d$ increase. This observation is consistent with the computational complexity of $\mathbf{SW}_{2,L}$ recalled in Section 2.1. On the other hand, our approximation scheme is extremely efficient even for large $d$ and $n$, since it is based on a simple deterministic formula which does not require projecting and sorting data along random directions.

**Image generation.** Finally, we leverage our theoretical insights to design a novel method for a typical generative modeling application. The problem consists in tuning a neural network that takes as input $k$-dimensional samples from a reference distribution (e.g., uniform or Gaussian), to generate images of dimension $d > k$. During the training phase, the parameters of the network are updated by iteratively minimizing a dissimilarity measure between the dataset to fit and the generated images.

In [9], the dissimilarity measure is Monte Carlo SW approximated with $10^4$ random projections, and the resulting generative model is called the *Sliced-Wasserstein generator* (SWG). This model performs well on moderately high-dimensional image datasets (e.g., $28 \times 28$ for MNIST images [42]). However, for very large dimensions (e.g., $64 \times 64 \times 3$ for the CelebA dataset [43]), Monte Carlo SW requires more than $10^4$ random projections to capture relevant information, which leads to very expensive training iterations and potential memory issues. To offer better scalability, SWG can be augmented with a discriminator network [9, Section 3.2] that aims at finding a lower-dimensional space in which the two projected datasets are clearly distinguishable. The intuition behind this heuristic is that the more distinct the two datasets are from each other, the fewer projection directions Monte Carlo SW requires to provide useful information. The training then consists in optimizing the generator's and discriminator's objective functions in an alternating fashion.

Our novel approach builds on SWG and modifies the saddle-point problem in [9, Section 3.2]: motivated by the gain in accuracy and time illustrated in Figure 2 on high-dimensional datasets, we propose to replace Monte Carlo SW with our approximate SW (14) in the generator's objective; then, to make sure that our approximation is accurate, we regularize the discriminator's objective:

$$\max_{\psi}\ L(\psi) + \lambda_1 \big\|\mathrm{Cov}[d'_\psi(X)]\big\|_F^2 + \lambda_1 \big\|\mathrm{Cov}[d'_\psi(g_\phi(Z))]\big\|_F^2 \tag{16}$$

$$+ \lambda_2\, \mathbb{E}\left[\|d'_\psi(X)\|^{-2}\right] + \lambda_2\, \mathbb{E}\left[\|d'_\psi(g_\phi(Z))\|^{-2}\right] \tag{17}$$

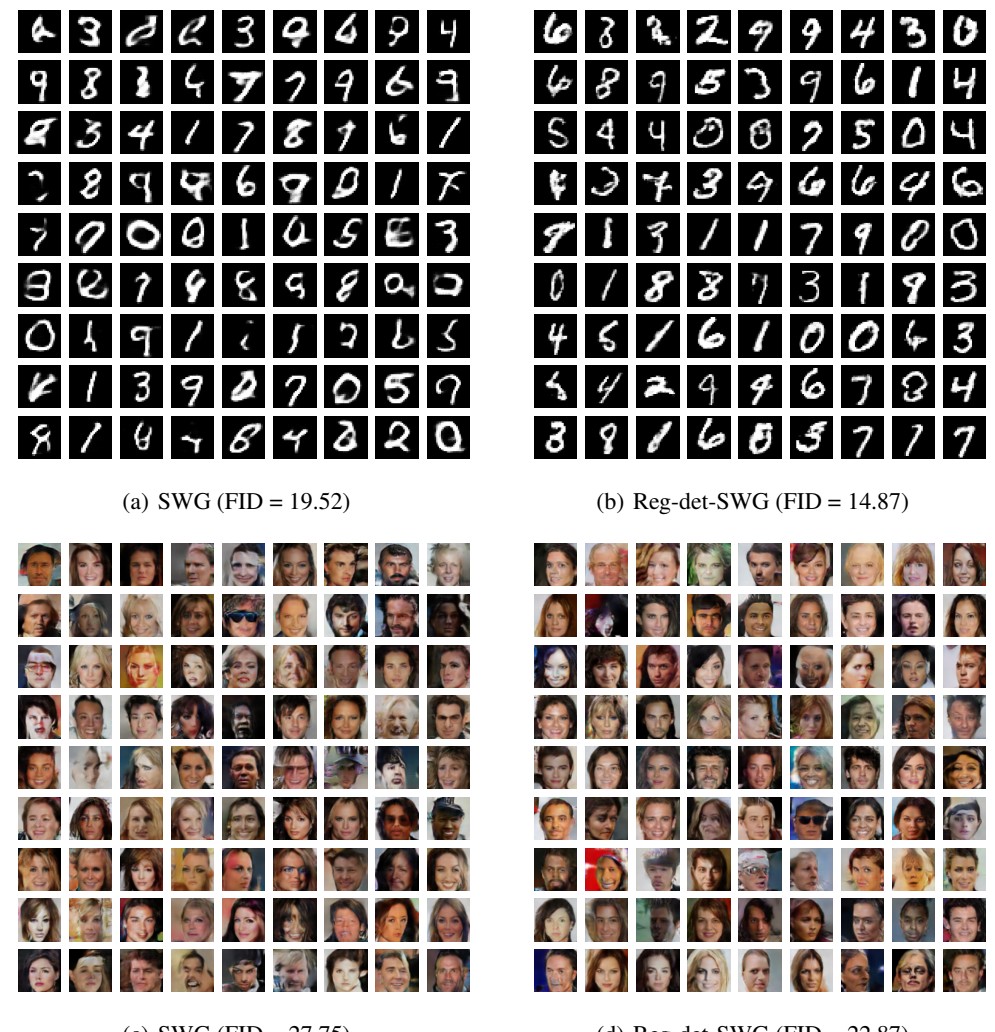

(a) SWG (FID = 19.52)

(b) Reg-det-SWG (FID = 14.87)

(c) SWG (FID = 27.75)

(d) Reg-det-SWG (FID = 22.87)

Figure 3: Images generated after training on MNIST (top row) and CelebA (bottom row). For each model, the images are associated with the lowest FID obtained over 5 runs.

where $L$ is the discriminator's loss used in SWG, $g_\phi$ and $d'_\psi$ are the generator's last layer and the discriminator's penultimate layer respectively (parameterized by $\phi$, $\psi$), $X$ and $Z$ are the random variables corresponding to the images to fit and the generator's input, Cov denotes the covariance matrix, $\| \cdot \|_F$ the Frobenius norm, and $\lambda_1, \lambda_2 \geq 0$. The regularization in (16) enforces the weak dependence condition (Corollary 1), while (17) prevents the network to converge to $d'_\psi = 0$. We call this generative adversarial network *regularized deterministic SWG* (reg-det-SWG).

To investigate the consequences of (i) regularizing the discriminator, and (ii) replacing the Monte Carlo SW with our approximation, we design another model, called *regularized SWG* (reg-SWG): similarly to SWG, the generator minimizes $\mathbf{SW}_{2,10^4}$, but the discriminator's objective is regularized as in (16), (17). We then compare reg-det-SWG against SWG and reg-SWG, by training the models on MNIST and CelebA and measuring their respective training time and Fréchet Inception Distances (FID, [44]): see Table 1. We used the same network architectures for all methods, and tuned $(\lambda_1, \lambda_2)$ via cross-validation: more details on the experimental setup are given in Section S4. First, we observe that the regularized models produce images of higher quality, since reg-SWG and reg-det-SWG return lower FID values than SWG. The FID of reg-SWG and reg-det-SWG are close for both datasets, thus the two models seem to yield similar performances. Hence, we report in Figure 3 the images generated by SWG and reg-det-SWG only.

The training process is more expensive when regularizing the discriminator: the average running time per epoch is higher for the regularized models. We also observe that reg-det-SWG is faster than reg-SWG, which is consistent with the fact that our approximation method is faster than Monte Carlo on high-dimensional settings. To further illustrate this point, we reported the average time spent in computing the generative loss per epoch, *i.e.* $\mathbf{SW}_{2,10^4}$ for SWG and reg-SWG, and $\widehat{\mathbf{SW}}_2$ for reg-det-SWG: see column $T_{\mathrm{SW}}$ in Table 1. On GPU, reg-det-SWG is at least 15 times faster than SWG and reg-SWG on MNIST, and 6 times faster on CelebA. Note that the models were trained using PyTorch, thus Monte Carlo SW benefits from a GPU-accelerated implementation of the sorting operation (with the function `torch.sort`). We also reported the computation times when models are trained on CPU. In this case, computing $\widehat{\mathbf{SW}}_2$ takes at most less than 3s per epoch, whereas the Monte Carlo estimation executes in several minutes (e.g., approximately 45min on CelebA). As a result, the total training time is almost the same for reg-det-SWG and SWG on CelebA, and the lowest for reg-det-SWG on MNIST.

## 5 Conclusion

In this work, we presented a novel method to approximate the Sliced-Wasserstein distance of order 2, which relies on the concentration-of-measure phenomenon for random projections. The resulting method computes SW with simple deterministic operations, which are computationally efficient even on high-dimensional settings and do not require any hyperparameters. We proved nonasymptotical guarantees showing that, under a weak dependence condition, the approximation error goes to zero as the dimension increases. Our theoretical findings are then illustrated with experiments on synthetic datasets. Motivated by the computational efficiency and accuracy of our approximate SW, we finally designed a novel approach for image generation that leverages our theoretical insights. As compared to generative models based on SW estimated with Monte Carlo, our framework produces images of higher quality with further computational benefits. This encourages the use of our approximate SW on other algorithms that rely on Monte Carlo SW, e.g. autoencoders [8] or normalizing flows [12].

The weak dependence condition can be inappropriate to describe the underlying geometry of real data in ML applications, and in that case, approximating SW with our method seems inadequate. To overcome this problem, we encourage practitioners to resort to models where real data are represented by features that can be made weakly dependent. This strategy has proven successful in our image generation experiment: the reg-det-SWG model uses our approximation to compare two sets of features (instead of the raw images) whose covariance matrices are regularized to enforce weak dependence. Since many ML techniques make use of features and regularizers, we believe that our methodology is not restrictive and can then be applied to other standard problems than image generation. Besides, our weak dependence condition in Definition 1 is weaker than the one in [39], which is a notion commonly used in statistics.

Our empirical results on synthetic data show that the approximation error goes to zero with a faster convergence rate than the one we proved. Then, the main current limitation of our framework is that our theoretical convergence rate in $d^{-1/8}$ might be slower than necessary. We proved that the overall approximation error is upper-bounded by a term in $d^{-1/2}$ when comparing Gaussians with diagonal covariance matrices, and the improvement of our error bounds for other specific distributions is left for future work. On the other hand, the extension of our methodology to variants of SW is another challenging future research direction. To the best of our knowledge, the literature on the concentration of measure phenomenon focuses on linear random projections, therefore the derivation of deterministic approximations for SW based on nonlinear projections seems highly nontrivial. A more promising direction would be to generalize our approach to SW based on $k$-dimensional linear projection by leveraging the bound in [26, Theorem 1] for $k > 1$.

Since this paper is focused on developing a theoretically-grounded novel method to estimate a distance between probability distributions, we believe it will not pose any negative societal or ethical consequence. On the other hand, as demonstrated in Section 4, our contribution provides tools to speed up existing machine learning algorithms on CPU, which is useful when powerful hardware resources are not available, or when their use is deliberately avoided for environmental purposes.

## Acknowledgments and Disclosure of Funding

This work is partly supported by the industrial chair "Data Science & Artificial Intelligence for Digitalized Industry & Services" from Télécom Paris. Umut Şimşekli's research is supported by the French government under management of Agence Nationale de la Recherche as part of the "Investissements d'avenir" program, reference ANR-19-P3IA-0001 (PRAIRIE 3IA Institute). Pierre E. Jacob gratefully acknowledges support by the National Science Foundation through grant DMS-1844695. Alain Durmus acknowledges support of the Lagrange Mathematical and Computing Research Center. Kimia Nadjahi is grateful to Pierre Colombo for his helpful advice on how to train the generative models in Section 4 on Télécom Paris's GPUs.

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
