# OpenReview forum: "Fast Approximation of the Sliced-Wasserstein Distance Using Concentration of Random Projections"
_NeurIPS.cc/2021/Conference — NeurIPS 2021 Poster_

### Official Review · Reviewer_QHXY · 2021-06-29

**Rating:** 6
**Confidence:** 4

**Summary:**

The paper proposes a new approach to approximate the sliced Wasserstein distance. In particular, the paper extends the central limit theorems for random projections to derive a deterministic approximation of sliced Wasserstein (based on the closed form of Wasserstein distance between two Gaussians) that is efficient (low error) in high dimension under centering assumption (weak dependence assumption). On the experimental side, the paper illustrates the derived theory on the synthetic data and compares the proposed approach to the conventional Monte Carlo approximation of the sliced Wasserstein distance.

**Limitations And Societal Impact:**

The weak dependence assumption is not easy to be satisfied in practice, additional experiments on other datasets  (CIFAR-10, LSUN) and other applications (e.g. autoencoders, approximate Bayesian computation) are needed. The approximation seems only work well with high dimensional measures, hence, I think it is necessary to do some experiments on applications that involves low dimensional  measures (latent space of autoencoders, color transfer).

Minor comments: FID scores should also be reported in the middle of the training process (different epochs).

Questions:
1. is it possible to extend the result to derive deterministic approximation for the non-linear projecting (generalized sliced Wasserstein) and other extensions of sliced Wasserstein such as max sliced Wasserstein, k-max sliced Wasserstein and spherical sliced Wasserstein [2]?
2. Can we derive an algorithm to transport a measure to a measure like the iterative distribution transfer algorithm for this type of approximation?

[2] Improving Relational Regularized Autoencoders with Spherical Sliced Fused Gromov Wasserstein

**Main Review:**

Originality: Gaussian projections were also used in the sliced score matching technique [1], however, the application to the sliced Wasserstein in the paper is original. The deterministic approximation of the sliced Wasserstein that is based on the usage of the central limit theorems for random projections is new and novel.

Quality: The submission is technically sound. The error between the proposed approximation and the true value of the sliced Wasserstein is bounded for both two assumptions. Moreover, the paper haves demonstrated the superiority of the new approximation over the Monte Carlo estimation in terms of both computation complexity and performance in generative modeling experiments.  The authors have also discussed the limitations of the approximation due to the independence (the weak dependence) assumption. The experiments are adequate for showing the faster computational speed of the deterministic sliced Wasserstein.

Clarity: The paper is well-written, easy to follow. The experimental settings are reported enough to reproduce. The proofs are written rigorously and correctly.

Significance: The paper pushes forward the practical applications of the sliced Wasserstein distance, and provides a new insight to understand the distance. However, the convergence rate is slow, hence, the approximation seems only efficient in the case of very high dimensional measures.

[1] Sliced Score Matching: A Scalable Approach to Density and Score Estimation

**Time Spent Reviewing:**

16

---

> ### Author Response · Authors · 2021-08-10
> **Response to Reviewer QHXY**
>
> We thank the reviewer for their encouraging and positive feedback. We appreciate that they find our submission novel, technically sound and well written, our experiments adequate, and our proofs rigorously and correctly written.
>
> **Convergence rate:** We agree with the reviewer that the convergence rate of our approximate SW seems slow according to our theoretical guarantees, while it is reasonably fast in our experiments. This suggests that there is room for improvement for our nonasymptotical guarantees, as we pointed out in lines 262-263 and 339-342. To bridge this gap between theory and practice, we further analyzed the rate: we considered the case where data are sampled from centered Gaussians with zero means and diagonal covariance matrices. By using the closed form expression of SW that we derived in Section S3.1 (supplementary document), we can show that the overall approximation error is upper-bounded by a term in $d^{-1/2}$, which is consistent with our synthetic experiments: see Figure 1(b) and line 261. Therefore, we will add this additional result and its complete proof to our paper, and we believe that it is a first encouraging step towards the following research direction: we will study if our proofs and the ones in [R5] can be refined when assuming additional structure on the distributions (e.g., sub-Gaussian and sub-exponential), in order to identify the settings under which our current bounds are tight or can be improved.
>
> **Additional experiments:** We will report the FID scores of our image generation experiment for intermediate epochs and add more empirical results to our paper, as suggested by the reviewer. Specifically, we will evaluate our generative models on CIFAR-10 and LSUN, and conduct an experiment for approximate Bayesian computation on time series (inspired by [Section 4, R1]).
>
> The reason why we focused on high-dimensional empirical settings is to be consistent with our theoretical results, which encourage the use of our methodology for sufficiently high values of $d$. Nevertheless, we agree with the reviewer that investigating the performance on low-dimensional measures would be interesting, and our experiments already provide promising results in that direction: our approximation is reasonably accurate for low and medium dimensions on the synthetic settings (Figures 1 and 2), and the features used for MNIST in the image generation experiment are of dimension 256, which is considerably small. As future work, we thus plan to further explain the behavior of our approximation on lower-dimensional settings by conducting adequate experiments and studying if the convergence rate can be sharpened under specific assumptions.
>
> **Extensions:** The extension of our methodology to variants of SW is a very interesting and challenging future research direction. To the best of our knowledge, the literature on the concentration of measure phenomenon focuses on linear random projections (see Section 2.2), therefore the derivation of deterministic approximations for SW based on nonlinear projections seems highly nontrivial.
>
> Regarding maximum SW distances, the contributions in [R2] might be useful, specifically the bound on the “maximum-sliced distance” between a distribution and its Gaussian approximation [Theorem 2, R2]: one could explore if this result could be extended to max SW. Finally, a promising direction would be to generalize our approach to SW based on $k$-dimensional linear projection with $k > 1$, by leveraging theoretical results from [R3,R4,R5]. We will add a discussion on this matter.
>
> We think it is possible to develop an algorithm to transport one measure to another using our approximation, since prior work proposed such methods based on the Monte Carlo approximation of SW (e.g., [R6, R7]), and we showed in our image generation experiment that it is possible to build new algorithms by incorporating our methodology in Monte Carlo SW-based methods. However, adapting the iterative distribution transfer method seems quite difficult: this technique aims at fitting a density by iteratively matching a finite number of its one-dimensional representations corresponding to projections, therefore the Monte Carlo approximation of SW can be incorporated more easily in this framework, as demonstrated in [R7]. We thank the reviewer for this interesting question, we will mention this aspect in the conclusion.
>
> [R1] “Approximate Bayesian computation with the Wasserstein distance”, Bernton et al. (2019)
>
> [R2] “The Gaussian equivalence of generative models for learning with shallow neural networks”, Goldt et al. (2021)
>
> [R3] “Approximation of projections of random vectors”, Meckes (2010)
>
> [R4] “Projections of probability distributions: A measure-theoretic
> Dvoretzky theorem”, Meckes (2012)
>
> [R5] “Conditional Central Limit Theorems for Gaussian Projections”, Reeves (2017)
>
> [R6] “Sliced-Wasserstein Flows: Nonparametric Generative Modeling via Optimal Transport and Diffusions”, Liutkus et al. (2019)
>
> [R7] “Sliced Wasserstein Generative Models”, Wu et al. (2019)

---

### Official Review · Reviewer_L6sj · 2021-07-11

**Rating:** 7
**Confidence:** 3

**Summary:**

This paper develops a novel deterministic approximation for the Sliced-Wasserstein Distance (SWD), based on the fact that a random linear projection of a high-dimensional vector has an approximately Gaussian distribution. In many cases, the approximation involves no tuning parameters and can be computed much more quickly than the usual Monte Carlo estimate of SWD, especially for very high-dimensional data. If the $d$ dimensions of the data are not too strongly correlated, then the approximation converges no slower than $d^{-1/8}$ as $d \to \infty$. Experiments with synthetic data validate this theoretical result, as well as the computational benefits of the proposed approximation. Finally, the paper demonstrates an application to image generation.

**Limitations And Societal Impact:**

The paper briefly discusses an apparent gap between the main theoretical and empirical results. I don't foresee any potential negative societal impact of the work.

**Main Review:**

Overall, the paper is very clearly written and well-motivated, and the proposed estimator seems novel and useful. From a statistical perspective, I have some concerns about the usefulness of the provided theoretical results.

Main Comments/Questions:
1) The paper bounds the error between the true SWD and the approximation when the true data distributions are used. In reality, empirical distributions over IID data, rather than the true distributions are used. Roughly speaking, the paper bounds the bias of the proposed estimator, but not the variance. Could the authors discuss the rate as which the variance decays as the sample size increases?

2) A convergence rate of d^{-1/8} is quite slow (a decrease by a factor of ~1.3 for each 10-fold increase in dimension). On the other hand, based on the synthetic experiments, it appears that the estimator might converge at a faster rate in practice. Could the authors speculate on whether this theoretical rate is tight (in the worst case) under the given fourth-order weak dependence assumption, and/or under what conditions the convergence rate might be faster? I see that this is noted in Lines 339-342, but it would be nice if the authors could expand on this a bit.

Minor Comments:
1) Line 126: Typo: "whose contributions has" -> "whose contributions have"

2) Line 128, "$\mu_d^X$, the distribution of $X_{1:d}$":  Upon first reading, it wasn't immediately clear to me whether $\mu_d^X$ was the *empirical distribution* of the data, the *population distribution* from which each X_i was drawn IID, or the *joint population distribution* of all d. If I understand correctly, the assumption that $\mu_d^X \in \mathcal{P}_2(\mathbb{R}^d)$ (on Line 129) only really makes sense under the last interpretation, but this distinction could be clearer on Line 128.

**Time Spent Reviewing:**

4

---

> ### Author Response · Authors · 2021-08-10
> **Response to Reviewer L6sj**
>
> We thank the reviewer for the intriguing questions and positive feedback. We are glad that they find our paper very clearly written and well-motivated, and our contribution novel and useful.
>
>  We will fix the typo and ambiguous notation mentioned in “Minor Comments”.
>
> Regarding the main questions,
>
> 1. This point is indeed important, and we can establish the convergence rate requested by the reviewer as follows. Let $\mu, \nu$ be two probability measures on $\mathbb{R}^d$ satisfying the assumptions of Corollary 3. Denote by $\hat{\mu}\_n = n^{-1} \sum_{i=1}^n \delta\_{x^{(i)}}$ and $\hat{\nu}\_n = n^{-1} \sum_{i=1}^n \delta_{y^{(i)}}$ their respective empirical approximations, where $(x^{(i)} )\_{i=1}^n$ and $( y^{(i)} )\_{i=1}^n$ are i.i.d. from $\mu$ and $\nu$ respectively. Then, by the triangle inequality and the linearity of the expectation,
> $$\mathbb{E} | \mathbf{SW}_2(\hat{\mu}_n, \hat{\nu}_n) - \mathbf{W}_2\{ \mathrm{N}(0, \mathrm{m}_2(\hat{\mu}_n)), \mathrm{N}(0, \mathrm{m}_2(\hat{\nu}_n)) \} | $$
> $$ \leq \mathbb{E} | \mathbf{SW}_2(\hat{\mu}_n, \hat{\nu}_n) - \mathbf{SW}_2(\mu, \nu) | $$
> $$ \quad + | \mathbf{SW}_2(\mu, \nu) - \mathbf{W}_2\{ \mathrm{N}(0, \mathrm{m}_2(\mu)), \mathrm{N}(0, \mathrm{m}_2(\nu)) \} | $$
> $$ \quad + \mathbb{E} | \mathbf{W}_2\{ \mathrm{N}(0, \mathrm{m}_2(\mu)), \mathrm{N}(0, \mathrm{m}_2(\nu)) \} - \mathbf{W}_2\{ \mathrm{N}(0, \mathrm{m}_2(\hat{\mu}_n)), \mathrm{N}(0, \mathrm{m}_2(\hat{\nu}_n)) \} | $$
> By the sample complexity of SW, the first term converges to zero as $n$ goes to infinity, and the convergence rate does not depend on the dimension $d$ (see [Corollary 2, R1] for the explicit rate). The second term, which does not depend on $n$, can be bounded by Corollary 1. Finally, the third term corresponds to a Monte Carlo error, since
> $$ \mathbb{E} | \mathbf{W}_2\{ \mathrm{N}(0, \mathrm{m}_2(\mu)), \mathrm{N}(0, \mathrm{m}_2(\nu)) \} - \mathbf{W}_2\{ \mathrm{N}(0, \mathrm{m}_2(\hat{\mu}_n)), \mathrm{N}(0, \mathrm{m}_2(\hat{\nu}_n)) \} | $$
> $$ = \mathbb{E} | \mathrm{m}_2(\mu)^{1/2} - \mathrm{m}_2(\nu)^{1/2} - (\mathrm{m}_2(\hat{\mu}_n)^{1/2} - \mathrm{m}_2(\hat{\nu}_n)^{1/2}) | $$
> $$ \leq \mathbb{E} | \mathrm{m}_2(\hat{\mu}_n)^{1/2} - \mathrm{m}_2(\mu)^{1/2} | +  \mathbb{E} | \mathrm{m}_2(\hat{\nu}_n)^{1/2} - \mathrm{m}_2(\nu)^{1/2} |\ , $$
> where for $\xi \in \\{\mu, \nu\\}$, $\mathrm{m}_2(\hat{\xi}_n)$ is the Monte Carlo approximation of $\mathrm{m}_2(\xi)$ (as defined in lines 181-184). Therefore, this third term goes to 0 with a convergence rate of order $n^{-1/2}$. We will add this proof and the explicit convergence rate to our manuscript.
>
> 2. As requested by the reviewer, we further analyzed the convergence rate to better explain the gap between theory and practice. We considered the case where data are sampled from centered Gaussians with zero means and diagonal covariance matrices. By using the closed form expression of SW that we derived in Section S3.1 (supplementary document), we can show that the overall approximation error is upper-bounded by a term in $d^{-1/2}$, which is consistent with our synthetic experiments: see Figure 1(b) and line 261. Therefore, we will add this additional result and its complete proof to our paper, and we believe that it is a first encouraging step towards the following research direction: we will study if our proofs and the ones in [R2] can be refined when assuming additional structure on the distributions (e.g., sub-Gaussian and sub-exponential), in order to identify the settings under which our current bounds are tight or can be improved.
>
> [R1] “Statistical and Topological Properties of Sliced Probability Divergences”, Nadjahi et al. (2020)
>
> [R2] “Conditional Central Limit Theorems for Gaussian Projections”, Reeves (2017)

---

> > ### Comment · Reviewer_L6sj · 2021-09-10
> > **Thanks to the authors for their response**
> >
> > Thanks to the authors for their detailed response, which suitably addressed my questions. Given this, I am raising my score from 6 to 7.

---

> > > ### Author Response · Authors · 2021-09-13
> > > **Thanks**
> > >
> > > We are very glad that our responses have addressed Reviewer L6sj's concerns. We will implement all the suggested improvements in our revision.

---

### Official Review · Reviewer_sFFp · 2021-07-16

**Rating:** 6
**Confidence:** 4

**Summary:**

The paper presents a method to estimate the Sliced-Wasserstein distance. The method is based on the Gaussian approximation of the projected data, thus the SW distance could be approximated through the SW distance between Gaussians which has a closed-form. Hence, the method is better than the Monte Carlo SW distance in term of computational speed. The authors also demonstrate their method on toy data set and real data sets, i.e. MNIST and Celeb A.

**Ethical Concerns:**

It is fine.

**Limitations And Societal Impact:**

It is fine.

**Main Review:**

The paper starts with reviewing some results from Reeves that the SW-2 distance between the  projected distribution and the corresponding Gaussian could be bounded by a constants which depends on the mean and covariance matrix. Then in Section 3,  it moves to the SW-2 distance with Gaussian projections, where they proved that  the SW-2 distance using unit projection vector is proportional to the SW-2 distance using Gaussian projection. This result is not surprising, since the way to generate random unit vector from uniform distribution on the sphere based on normalzing a Gaussian vector. From the theoretical point of view, normalzing a Gaussian vector in high dimensional space to form an unit vector will not affect too much to the effect of the SW-2 distance. Because in high dimensional space, a Gaussian vector with independent components has a similar magnitude of length by the law of large number.  Using Gaussian projection might only  have some advantages of playing with the formulas rather than the effectiveness of the projection.

In section 3.2,  the authors  provide a similar bound for the difference between the SW-2 distance between two probability measures and W-2 distance between the corresponding Gaussian distributions, which is bounded by function of covariance matrices.  Section 3.3 shows the error analysis under the assumption of weak dependence between vector's components and the dependency between vector's component decrease as the gap between components increases.  The reviewer personally finds the condition of weakly dependency quite unrealistic.  It could be realistic in some cases like time series data. But weak dependency is closer to independency, thus the point cloud of the distribution is all over the space, i.e point cloud of Gaussian distribution. It  is quite contradictory to the popular view that the data generally lie in a low dimensional manifold. What could the authors comment about it?

The experiment section starts with a toy example of synthetic data, it is time series  data, so it is quite different from other data sets, because there is an order between components and weakly dependency condition over the order of components. For the real data set, they present 3 methods of  two using the Monte Carlo.  Among them, the Reg-det-SWG obtains a comparable FID score but a much faster processing time. However, the authors have not compared their methods with other methods of selecting important directions for projection.

Overall, this work is based on approximating the SW-2 distance. The main question is that why on the real data the approximation produces a comparable performance  with  the Monte Carlo method even when the number of the projection for the Monte Carlo method is large and the upper bound for the approximation error is not small? Because of the above-mentioned reasons, I do not believe that the theory of weak dependency is the  explanation for the comparable performance of the method on the real data set. Hence, there needs at least an ablation study to shed some light on the question.

**Time Spent Reviewing:**

8 hours

---

> ### Author Response · Authors · 2021-08-10
> **Response to Reviewer sFFp**
>
> We thank the reviewer for their insightful comments and the positive feedback.
>
> **Weak dependence condition:** We agree with the reviewer that the weak dependence condition can be inappropriate to describe the underlying geometry of real data in machine learning (ML) applications, and in that case, approximating SW with our method seems inadequate. To overcome this problem, we encourage practitioners to resort to models where real data are represented by features that can be made weakly dependent. This strategy has proven successful in our image generation experiment: the “Reg-det-SWG” model uses our approximation to compare two sets of features (instead of the raw images) whose covariance matrices are regularized to enforce weak dependence. Since many machine learning techniques make use of features and regularizers, we believe that our methodology is not restrictive and can then be applied to other standard problems than image generation, which we will develop as future work. On the other hand, our weak dependence condition (Definition 1) is weaker than the one in [R1], which is a notion commonly used in statistics. The reviewer is right in examining the practical implications of weak dependence, which is an important aspect of our study, so we will add this discussion to our paper.
>
> **Empirical comparison:** The comparison of our approach with “other methods of selecting important directions for projection” (e.g., max-SW [R2] and generalized SW [R3-5]) would indeed be interesting, and we will evaluate their performance (in terms of quality and computational time) on our empirical settings in Section 4.
>
> However, we believe that our proposal is quite different from this line of work for two main reasons: first, our method directly approximates SW (thus, does not define an alternative distance) and comes with theoretical guarantees on its induced error. Then, our approximation is computed via simple deterministic operations and does not rely on any hyperparameters, as opposed to the aforementioned variants of SW: these are defined as optimization problems and are computed by finding an approximate solution, which incurs an additional computational cost and estimation error.
>
> **Performance on real data:** As suggested by the reviewer, we conducted an ablation study on the image generation problem to clarify why the use of our approximation yields a comparable or better performance (in terms of FID). The architecture of the generative model based on our approximation (“Reg-det-SWG”) is driven by our theoretical findings (Corollary 1): we train a generative adversarial network that uses our approximate SW to compare features of images. These features are learned and enforced to be weakly dependent by using a discriminator network with a specific regularized objective (see equations (16) and (17)). Therefore, our ablation study consists in evaluating the performance of two additional generative models, which are respectively obtained by removing the discriminator in “SWG” (or equivalently, “Reg-SWG”) and “Reg-det-SWG”. In other words, each of these models is a generator which minimizes Monte Carlo SW (“SWG without disc.”) or our approximation (“Det-SWG”) between sets of raw images. On MNIST, we obtained an average FID score of $23.89 \pm 4.37$ for “SWG without disc.”,  and $282.6 \pm 32.7$ for “Det-SWG” (over 5 runs). Hence, “Det-SWG” performs significantly worse than the other models (Table 1). Since real image data are not weakly dependent (see lines 201-202), we think this supplementary experiment is in line with our intuition regarding the importance of weak dependence. We will add this discussion and the new empirical results (quantitative and qualitative) to our manuscript.
>
> [R1] “Probability and moment inequalities for sums of weakly dependent random variables, with applications”, Doukhan and Neumann (2007).
>
> [R2] “Max-Sliced Wasserstein distance and its use for GANs”, Deshpande et al. (2019)
>
> [R3] “Generalized Sliced Wasserstein Distances”, Kolouri et al. (2019)
>
> [R4] “Augmented Sliced Wasserstein Distances”, Chen et al. (2020)
>
> [R5] "Distributional Sliced-Wasserstein and Applications to Generative Modeling", Nguyen et al. (2021)

---

> > ### Comment · Reviewer_sFFp · 2021-09-11
> > **Reply**
> >
> > I would like to thank the authors for their response. I would like to keep my score unchanged, because I think the theory could not explain fully the whole picture of this approach.

---

> > > ### Author Response · Authors · 2021-09-13
> > > **Thanks**
> > >
> > > We thank Reviewer sFFp for considering our response letter and keeping their positive score. We would still be very happy to clarify if there are any specific questions about our response.

---

### Official Review · Reviewer_Zsjv · 2021-07-22

**Rating:** 7
**Confidence:** 4

**Summary:**

This paper introduces a (possibly) cheaper approximation of Sliced Wasserstein (SW) distance. The proposed technique is based on calculating 1D Wasserstein distance between Gaussian projection directions. This is motivated by the fact that Wasserstein distance between Gaussian distributions admits a closed-form solution, which leads to an easy compute of the proposed approximate SW and discard a large number of uniformly sampling random projections for the Monte Carlo approximation of the original SW distance.


**Limitations And Societal Impact:**

Yes

**Main Review:**

Strengths: the authors propose a new approach for approximate the SW distance using 1D Wasserstein distance between Gaussian distributions. Under mild conditions, the authors provide nonasymptotic guarantees on the error between the original and proposed approximation of SW distance. Overall, I find the exposure being well supported by the theory and the results are important to the field of sliced Wasserstein distances research direction. The paper is clearly written. The authors introduce all the material needed to understand the method. The proofs are well detailed and seem reasonable.

Weaknesses: the rate $d^{-1/8}$ for the nonasymptotical analysis seems (too) slow, whereas some experiments show a decay for this error for reasonable values of $d$. There is one important missing work that should be cited though:

Nguyen, Khai, et al. "Distributional Sliced-Wasserstein and Applications to Generative Modeling." (https://arxiv.org/abs/2002.07367)

where authors proposed the distributional slicing approach which is a general technique to design a probabilistic way to select important direction by finding optimal distribution over explored projections. I really believe this paper could also lead to interesting computational comparisons.



**Time Spent Reviewing:**

10

---

> ### Author Response · Authors · 2021-08-10
> **Response to Reviewer Zsjv**
>
> We thank the reviewer for their positive feedback, and finding our contributions well supported by the theory, important to the field of sliced Wasserstein distances, and clearly written. Our detailed responses are given below.
>
> **Convergence rate:** We agree with the reviewer that the convergence rate of our approximate SW seems slow according to our theoretical guarantees, while it is reasonably fast in our experiments. This suggests that there is room for improvement for our nonasymptotical guarantees, as we pointed out in lines 262-263 and 339-342. To bridge this gap between theory and practice, we further analyzed the rate: we considered the case where data are sampled from centered Gaussians with zero means and diagonal covariance matrices. By using the closed form expression of SW that we derived in Section S3.1 (supplementary document), we can show that the overall approximation error is upper-bounded by a term in $d^{-1/2}$, which is consistent with our synthetic experiments: see Figure 1(b) and line 261. Therefore, we will add this additional result and its complete proof to our paper, and we believe that it is a first encouraging step towards the following research direction: we will study if our proofs and the ones in [R1] can be refined when assuming additional structure on the distributions (e.g., sub-Gaussian and sub-exponential), in order to identify the settings under which our current bounds are tight or can be improved.
>
> **Additional reference:** We thank the reviewer for pointing out the following related work, [R2] "Distributional Sliced-Wasserstein and Applications to Generative Modeling." by Nguyen et al. (ICLR 2021). We agree that the proposed distributional Sliced-Wasserstein distance (DSW) seems relevant to our study, so we will add a discussion on how DSW compares with our approach. Specifically, DSW is defined as an optimization problem whose resolution is challenging, as pointed out in [R2, Section 3.2]. To overcome this issue, the authors propose to approximate the solution of the dual form by training a deep neural network. Hence, DSW is more computationally demanding than our approach, which relies on simple deterministic operations and does not need any hyperparameter tuning. In this sense, our contributions are rather orthogonal to theirs. Apart from this computational aspect, DSW has been shown to yield favorable results in generative modeling applications, therefore we will also evaluate its performance in our image generation experiment.
>
> [R1] “Conditional Central Limit Theorems for Gaussian Projections”, Reeves (2017)

---

> > ### Comment · Reviewer_Zsjv · 2021-09-11
> > **Thanks for the authors for their response**
> >
> > I would like to thank the authors for their response. The rebbutal addressed my concerns, so I decide to change my score from 6 to 7.

---

> > > ### Author Response · Authors · 2021-09-13
> > > **Thanks**
> > >
> > > We are very happy to hear that our responses have addressed Reviewer Zsjv's concerns. We are committed to making the suggested improvements in our revision.
> > >
> > > On another note, unfortunately we are not able to see a change in the overall score (it still seems 6). We would like to mention this point to prevent a potential technical issue.

---

### Decision · Program_Chairs · 2021-09-27

**Decision:**

Accept (Poster)

**Comment:**

The focus of the submission is the fast approximation of the sliced-Wasserstein distance (SW). Particularly, the authors present a new scheme to tackle this task relying on specifically designed Gaussian projections as an alternative to the widely-used Monte Carlo approximation (where the projection directions are distributed uniformly on the unit sphere and their number often has to be large to achieve highly-accurate approximation), with consistency guarantees. The efficiency of the approach is demonstrated on synthetic examples and in generative modelling (tuning a neural network for image generation with SW objective).

Estimating the discrepancy of probability measures in R^d is a fundamental problem in statistics and machine learning with numerous applications. The submission is a well-organized, clearly-written work in this direction where the authors deliver important tools (SW estimator) with sound theoretical guarantees as assessed by the reviewers; it can be of definite interest to the ML community.